# Influence of Chemical Pretreatment on the Mechanical, Chemical, and Interfacial Properties of 3D-Printed, Rice-Husk-Fiber-Reinforced Composites

Athira Nair Surendran [1,2], Sreesha Malayil [1,3], Jagannadh Satyavolu [1] and Kunal Kate [2,*]

1   Conn Center for Renewable Energy Research, University of Louisville, Louisville, KY 40208, USA; athiranair.surendran@louisville.edu (A.N.S.); smalayil@rti.org (S.M.); jagannadh.satyavolu@louisville.edu (J.S.)
2   Materials Innovation Guild, Department of Mechanical Engineering, University of Louisville, Louisville, KY 40208, USA
3   RTI International, Research Triangle Park, NC 27709, USA
*   Correspondence: kunal.kate@louisville.edu

**Abstract:** This article explores using biomass, namely rice husks, as a reinforcement material in thermoplastic copolyester (TPC) composites. Rice husks were subjected to three chemical pretreatments: single-stage sulfuric acid hydrolysis, first-stage sulfuric acid hydrolysis followed by a second-stage methanesulfonic acid (MSA) treatment, and first-stage sulfuric acid hydrolysis followed by a second-stage sodium hydroxide alkali treatment. We studied the effects of these treatments on the rheological, thermal, interfacial, and mechanical properties of composites. The fibers were mixed with polymers at high shear rates and temperatures, and 3D-printed filaments were produced using a desktop 3D printer. The printed parts were analyzed using tensile tests, torque and viscosity measurements, and thermogravimetric analysis to obtain their mechanical, rheological, and thermal properties. SEM imaging was performed to understand the fiber–polymer interface and how it affects the other properties. The results showed that first-stage sulfuric acid hydrolysis followed by a second-stage pretreatment of the fibers with MSA showed better fiber–polymer adhesion and a 20.4% increase in stress at 5% strain, a 30% increase in stress at 50% strain, and a 22.6% increase in the elastic modulus as compared to untreated rice husk composites. These findings indicate that readily available and inexpensive rice husks have significant potential for use in natural fiber-reinforced composites when pretreated using dilute sulfuric acid followed by methane sulfonic acid hydrolysis.

**Keywords:** rice husks; natural-fiber-reinforced composites (NFRCs); fused filament fabrication (FFF); thermoplastic copolyester (TPC); methanesulfonic acid pretreatment; dilute acid hydrolysis; mechanical properties; interfacial properties

## 1. Introduction

Additive manufacturing (AM) is an alternative to casting and injection molding, where materials can be added to form complex structures [1]. Also known as three-dimensional printing (3DP), AM has become more popular in the past ten years because of its flexibility, allowing for customization and reduced material wastage; it can be applied in fields such as the automotive industry [2], medicine [3,4], aerospace [5], and infrastructure construction [6,7]. Fused deposition modeling (FDM), also known as fused filament fabrication (FFF) or material extrusion, stands out as a prominent additive manufacturing method. In this process, successive layers are extruded onto a heated bed, gradually solidifying and culminating in a fully formed component. The appeal of FFF lies in its user-friendly nature, requiring minimal upkeep, while enabling the fabrication of intricate and elaborate models with negligible material loss. However, certain limitations persist, including suboptimal surface finish of the

produced parts, relatively slow production rates, and the potential for delamination between individual layers [8].

The challenges above contribute to the compromised mechanical characteristics of components produced via the FFF technique. A promising approach to enhance the 3D printed properties involves the integration of fiber composite filaments. By incorporating these specialized filaments, the resulting printed components exhibit markedly improved mechanical properties, addressing the shortcomings associated with traditional FFF printing [9]. When it comes to fiber-reinforced composites, two main types of fibers are used: synthetic and natural. Synthetic fibers, such as glass and carbon fibers, come in short and continuous forms and generally offer superior mechanical properties. On the other hand, natural fibers can be obtained from various sources, such as agricultural plant residue or industrial crops, and their use is becoming increasingly popular in composite research as it helps increase product sustainability. However, one of the biggest obstacles in creating natural-fiber-reinforced composites, particularly for AM, is the weak bond between the fiber and polymer matrix [10].

Chemical pretreatment has shown the potential to improve fiber–polymer bonding, and it can significantly enhance the mechanical properties of the composite. Bartnikowski et al. [11] examined the effects of hydrochloric acid and sodium hydroxide hydrolysis on the properties of 3D-printed poly($\varepsilon$-caprolactone) (PCL) scaffolds for biomedical applications. They investigated different concentrations of chemicals and hydrolysis reaction times. The results showed that exposure to sodium hydroxide caused surface degradation, significantly increasing surface charge and little effect on mechanical strength. At the same time, the hydrochloric acid treatment led to bulk degradation, resulting in a slight increase in surface charge and decreases in molecular weight and mechanical strength. Korniejenko et al. [12] investigated various concentrations of sodium hydroxide on fly ash and flax fiber molds, which were tested for geo-polymer applications after six months of storage. Their findings showed that the chemically pretreated fiber reinforced composites properties such as density, flexural, and compressive strength deteriorated. Surprisingly, these results did not match those from other studies using similar flax fibers methods. The reason behind this may be due to the Polish species of flax or the presence of other contaminants. On the other hand, Khosravani et al. [13] looked into the effects of post-processing acrylonitrile butadiene styrene (ABS) by using a chemical surface treatment of acetone on the printed part. The results revealed that the Young's Modulus, fracture load, and fracture toughness were reduced due to the breakdown of ABS bonds on the surface of the printed parts. However, open porosity decreased, and the water absorption coefficient increased, indicating that the part is more hydrophobic.

Another study by Chinga-Carrasco et al. [14] demonstrated how hydrothermal and soda-pulping treatments can enhance the properties of bagasse-based cellulose nanofibrils (CNF). This chemical pretreatment makes them suitable for 3D printing ink for biomedical devices. The researchers considered cytotoxicity, which means that the bio-inks need to pose little to no toxicity to be applied in the biomedical field. The study found that hydrothermally and soda-treated bagasse pulp CNF was noncytotoxic and the best pretreatment for 3D printing ink. Marichelvam et al. [15] used sodium hydroxide (NaOH) to chemically remove hemicellulose, lignin, and other fatty materials from palm sheaths and sugarcane bagasse. The researchers found that all the alkali-treated fiber composites showed better mechanical properties. In fact, 40%-treated fiber content exhibited a Young's Modulus 51.75% greater than that of the untreated fibers with the same percent loading. Balla et al. [16] also studied the properties of sulfuric-acid-treated soy hull fiber composites. The results showed that the chemically treated soy hull fibers composites exhibited about a 50% increase in the elastic modulus and 5% and 50% strain. This improvement was attributed to the enhanced fiber–polymer interface. Bharath et al. [17] fabricated biodegradable printed circuit boards (PCB) with 60% loading of epoxy resin and 40% loading of rice husks and found that their specimens could withstand a tensile loading of 7 kN and

a flexural loading of 5 kN, making them a suitable composite PCB. Although there has been significant research on chemical pretreatments for composite applications, there is a scarcity of studies on chemical pretreatments of rice husks for composite processing and their 3D printing capabilities. Moreover, using biomass in 3D printing composites reduces the percentage of plastic in the printed part, making it more sustainable and eco-friendly.

This study explored the potential use of rice husks as a reinforcement material in thermoplastic copolyester (TPC) composites. Adding rice husks to TPC composites can produce low-carbon intensity printed parts, which is excellent news for the environment! We utilized biomass residues, which are carbon neutral, to create these composites. To improve the interfacial bonding between fibers, we subjected the rice husks to chemical hydrolysis using three different chemicals: sulfuric acid, methane sulfonic acid, and sodium hydroxide. The results showed that the interfacial bonding between chemically treated fibers was better compared to untreated fibers. We also optimized the printing parameters using L9 Taguchi statistical analysis method to ensure good part quality for mechanical analysis. Finally, we subjected the composites to thermal, viscosity, and mechanical tests and SEM imaging to examine interfacial bonding. Our investigation aimed to determine whether a chemically treated rice husk can be a suitable composite for TPC and act as a reinforcement to provide better mechanical and fiber–polymer interfacial properties.

## 2. Materials and Methods

### 2.1. Production of TPC-Rice Husk Composites

Rice husks were sourced from Riceland, Arkansas, and subjected to chemical treatment to enhance their suitability as reinforcement in thermoplastic copolyester (TPC) composites. Four distinct sample groups were prepared, each representing a different treatment condition: (1) As-Received Rice Husks (AR-RH), (2) (Hydrolyzed Rice Husks (H-RH), (3) Hydrolyzed and MSA-Treated Rice Husks (H-MSA-RH) and (4) Hydrolyzed and Alkali-Treated Rice Husks (H-AL-RH). The AR-RH) sample group comprised untreated rice husks and served as the baseline reference, the H-RH sample group had the rice husks hydrolyzed in H2SO4 at 150 °C with a 6% dry basis loading for 60 min, the H-MSA-RH sample group used the H-RH and further treated with them with 10% methane sulfonic acid (MSA). Here, a fiber-to-liquid ratio of 1:10 was employed, and the fibers were subjected to a 60-min cooking process at 140 °C. Lastly, in the H-AL-RH sample group, the rice husks were hydrolyzed with H2SO4 and treated with sodium hydroxide (NaOH). The fiber-to-liquid ratio was maintained at 1:10, and the treatment was conducted at 100 °C for 60 min. All treated and untreated rice husk samples underwent grinding and sifting to achieve particle sizes below 200 μm. The resulting samples were then placed in an oven at 50 °C for 2 h to eliminate moisture, ensuring a moisture content of less than 5%. The fibers were subsequently used for composite formulation and processing to make feedstock and filaments. The compositions of the five different sample groups and their designations are outlined in Table 1. Notably, all fiber-loaded samples consistently loaded 10 wt.% rice husk fiber. To prepare the composites, a weight ratio of 1:9 between dried rice husk and TPC (Hytrel 4056, DuPont, Wilmington, DE, USA) was established to create a natural fiber-reinforced composite (NFRC) with a consistent 10 wt.% rice husk fiber loading. The composite formulation involved mixing rice husk and TPC in a torque rheometer (Intelli-Torque Plasti-Corder, C. W. Brabender Instruments, Inc., New Jersey, NJ, USA) at 40 rpm and 160 °C. The mixing duration spanned 10–12 min, carefully controlled to prevent fiber decomposition. Subsequently, the composite material was granulated and fed into a capillary rheometer (Rheograph 20, GÖTTFERT Werkstoff-Prüfmaschinen GmbH, Buchen, Germany) equipped with a ∅1.75 mm tungsten carbide die. Subsequently, after the feedstock preparation, the composite granules were extruded at 160 °C, employing an extrusion speed of 0.05 mm/s, yielding 3D printing filaments.

**Table 1.** Designation of prepared TPC composite samples.

| Sample Name | Type of Rice Husk | First Treatment | Second Treatment |
|---|---|---|---|
| TPC | None | N/A | N/A |
| AR-RH | As-received | Untreated | N/A |
| H-RH | Hydrolyzed | Sulfuric acid | N/A |
| H-MSA-RH | Hydrolyzed and MSA treated | Sulfuric acid | Methane sulfonic acid |
| H-AL-RH | Hydrolyzed and alkali treated | Sulfuric acid | Sodium hydroxide |

*2.2. Optimization of Parameters for 3D Printing*

In order to find the best 3D printing process conditions, we printed 10 mm × 10 mm × 5 mm cuboids using pure TPC using a desktop FFF 3D Printer (Pulse XE, MatterHackers, Lake Forest, CA, USA) under different conditions. To determine these conditions, we conducted a literature study, consulted the manufacturer's recommendation, and took into account the thermal behavior of the material. Table 2 displays the parameter combinations for nine experiments, where three print temperatures, print speeds, and layer heights were varied. We used the Taguchi L9 optimization method to generate main effects plots for surface roughness in four directions, density, and dimensions of the printed parts. We found that a bed temperature of 80 °C was necessary for good adhesion of the printed part to the bed, and we chose a nozzle diameter of 0.6 mm to allow for fiber reinforcements to be printed homogeneously and to avoid nozzle clogging. All the parts were printed with 100% infill to ensure the high strength of the printed parts. To generate G-codes for the 3D printers, we utilized MatterControl software.

**Table 2.** Printing parameters for L9 Taguchi statistical analysis.

| Experiment Number | Print Speed (mm/s) | Layer Height (mm) | Print Temperature (°C) |
|---|---|---|---|
| 1 | 15 | 0.2 | 220 |
| 2 | 15 | 0.25 | 230 |
| 3 | 15 | 0.3 | 210 |
| 4 | 20 | 0.2 | 230 |
| 5 | 20 | 0.25 | 210 |
| 6 | 20 | 0.3 | 220 |
| 7 | 25 | 0.2 | 210 |
| 8 | 25 | 0.25 | 220 |
| 9 | 25 | 0.3 | 230 |

*2.3. Thermal Behavior of Composites*

The composite granules were subjected to thermogravimetric analysis (TGA) and differential scanning calorimetry (DSC) experiments to identify the constituents of the composites and to determine their transition temperatures. The analysis was carried out in a nitrogen atmosphere using the SDT TA system from TA Instruments (New Castle, DE, USA), with a sample size of 10–15 mg. The temperature was increased to 600 °C at an increment of 10 °C/min.

*2.4. Rheology Study of Composites*

To determine whether the composites had mixed homogenously, mixing torque data from the Brabender machine were evaluated at the end of each compounding experiment. A stable mixing torque indicated a homogenous feedstock of rice husk fiber and TPC matrix. The time-dependent viscosity was measured at 160 °C and a speed of 0.05 mm/s for 12 min using a 1.75 mm tungsten carbide die, which revealed the extrusion viscosity. Meanwhile, the shear-dependent viscosity was measured at 220 °C and various shear rates of 10, 20, 40, 80, 160, 400, and 800 $s^{-1}$ using a 1 mm tungsten carbide die to analyze the behavior of the different composites under different shear rates. These measurements provide valuable insights into the flow properties of the composite material, which in turn influenced the 3D printing condition.

### 2.5. Surface Morphology of Printed Parts

The Apreo C Scanning electron microscopy (SEM) from Thermo Fisher in Waltham, MA, USA, was used to image the 3D printed part's top surface, side profile, and cross-section. Therefore, all the 3D-printed samples were first sectioned with a rotary blade to obtain various cross-sections within the part and subsequently cased in an epoxy resin mount and polished to obtain an even sample height and bring a detailed display of fiber distribution, porosity, and fiber–polymer interface.

### 2.6. Mechanical Properties of Printed Composite Parts

The ASTM D638 type IV standard was used to perform tensile testing on the 3D-printed tensile bars. Three tensile bars were 3D printed with the above standard and tested using a universal tensile test machine (Instron Series 5560 A, Norwood, MA, USA). We chose a crosshead speed of 100 mm/s, and the test was conducted according to the ASTM D638 guidelines. A Shore D Hardness Tester was used to determine the hardness of the printed parts, and measurements were performed in the middle of each part. Additionally, the average diameter of the filaments was measured using calipers.

## 3. Results

### 3.1. Design of Experiments

The study focused on analyzing printed cuboids with dimensions of 10 mm × 10 mm × 5 mm, and the analysis encompassed key parameters including print speed, layer height, and print temperature. Each parameter contributed two degrees of freedom, amounting to a total of 6 DOF. An additional degree of freedom was allocated to the consideration of errors, bringing the total degree of freedom to 8.

The aim of the analysis was to evaluate the primary effects on cuboid dimensions and surface roughness, with the goal of achieving nominal values for dimensions while minimizing surface roughness. The empirical observations revealed nominal measurements of 10.62 mm, 10.8 mm, and 5.12 mm for length, width, and height, respectively (Figure 1).

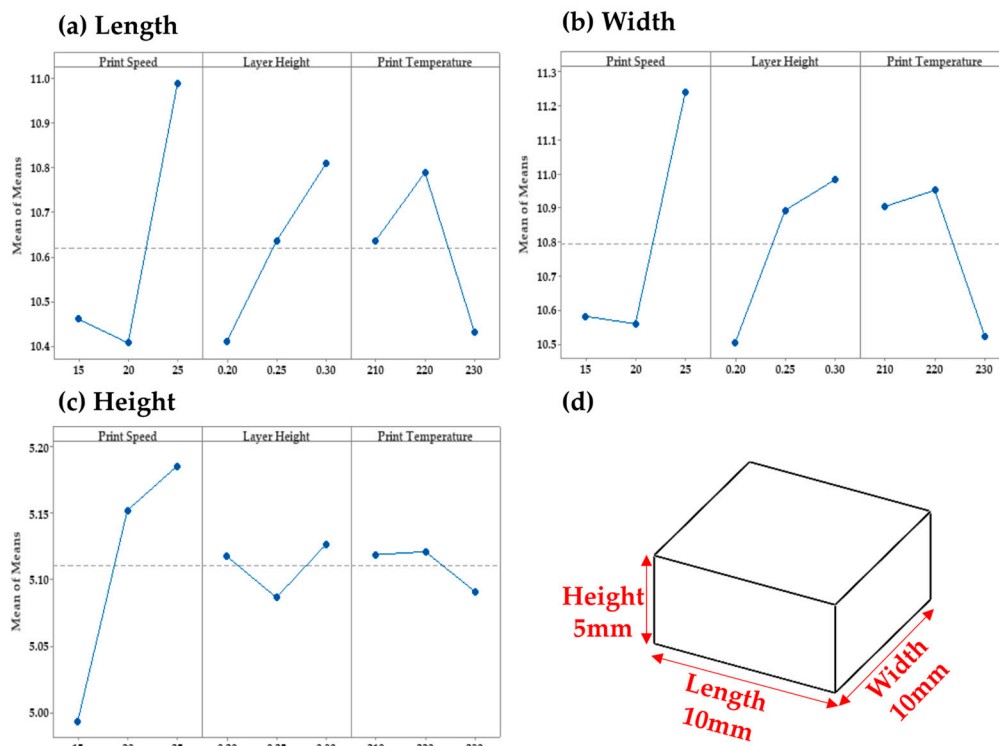

**Figure 1.** Effect of print speed, layer height, and print temperature on (**a**) length, (**b**) width, and (**c**) height of the printed part. (**d**) Representation of the printed part and dimensions.

Optimal print conditions were identified for achieving nominal dimensions, and distinctive patterns were observed. To achieve nominal length, a print speed of 15 mm/s, a layer height of 0.25 mm, and a print temperature of 210 °C were found to be effective. For nominal width, an optimal combination included a print speed of 20 mm/s, a layer height of 0.25 mm, and a print temperature of 210 °C. To achieve nominal height, an effective print configuration comprised a print speed of 20 mm/s, a layer height of 0.20 mm, and a print temperature of 210 °C.

The study graphically illustrates the optimal conditions for attaining nominal dimensions, highlighting the intricate interplay between print parameters and the resultant cuboid measurements. This systematic analysis provides valuable insights into achieving desired cuboid dimensions while emphasizing the critical influence of print speed, layer height, and print temperature on the quality and precision of printed parts.

When it comes to fused filament fabrication (FFF), achieving a smooth finish with low surface roughness is the top priority. Statistical analysis was conducted using a "minimum is best" approach to determine the best parameter settings. The Taguchi analysis revealed mean surface roughness values for different directions across and along the print direction. For instance, the sides across the print direction had a surface roughness of 1.95 μm, while the sides along the print direction had 11.4 μm (Figure 2).

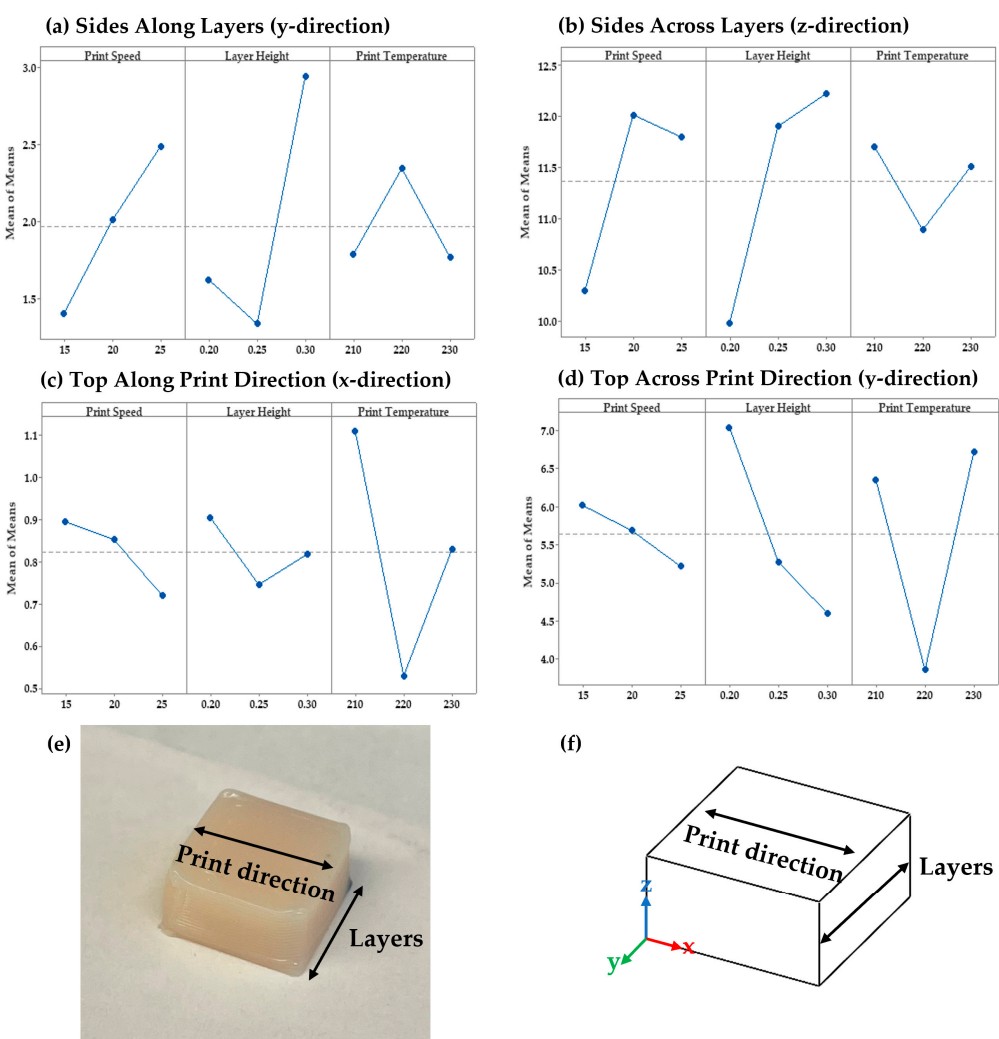

**Figure 2.** Effect of print speed, layer height, and print temperature on the surface roughness on the (**a**) y-direction on the sides, (**b**) z-direction on the sides, (**c**) x-direction on the top surface, and (**d**) y-direction on the top surface. (**e**) Prototype of printed part. (**f**) Representation of the surface roughness directions on printed parts.

To attain minimal surface roughness, a print speed of 15 mm/s, a layer height of 0.25 mm, and a print temperature of 230 °C were found to be optimal for the sides across layers. Meanwhile, the optimal conditions for achieving minimum surface roughness along layers were a print speed of 15 mm/s, a layer height of 0.2 mm, and a print temperature of 220 °C. Minimal surface roughness on the top surface was achieved with a print speed of 25 mm/s and a print temperature of 220 °C for both directions.

The optimal layer height settings for minimal surface roughness were 0.25 mm and 0.3 mm for surface roughness across and along the print direction on the top layer, respectively. The main effects plot for means and the ranking analysis converged on an optimal configuration for achieving the best dimensions and minimal surface roughness, which involves a print temperature of 220 °C, a print speed of 15 mm/s, and a layer height of 0.25 mm. Other print parameters were held constant, with notable selections including an 80 °C bed temperature, 100% infill density, and a grid infill pattern.

### 3.2. Thermal Behavior of Composites

The thermogravimetric analysis (TGA) of both pure TPC and rice husk composites is presented in Figure 3. The derivative weight loss percentage and weight loss data provide insights into the materials' thermal degradation behavior.

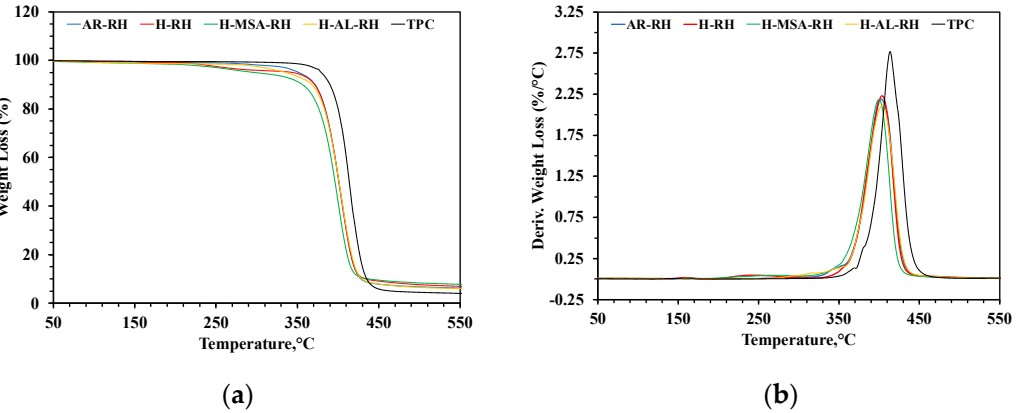

**(a)**          **(b)**

**Figure 3.** Thermogravimetric Analysis of pure TPC and rice husk fiber composites (**a**) Weight loss percentage and (**b**) derivative weight loss of rice husk composites.

The "onset point temperature" represents the initial temperature at which the degradation process commences [18].

The results reveal that among the composite samples, the H-AL-RH composite exhibited the lowest onset temperature at 337.7 °C, marking a decrease of 7.2% compared to pure TPC. This decline in onset temperature can be attributed to the incorporation of rice husks, which possess lower degradation points, as outlined in Table 3. Notably, second-stage-treated fibers displayed reduced levels of lignin, hemicellulose, and impurities, resulting in a lower onset degradation temperature. Importantly, these findings underscore the stability of all samples under filament extrusion and printing temperatures.

**Table 3.** Transition temperatures of TPC and its rice composites.

|  | Onset (°C) | Peak 1 (°C) | Peak 2 (°C) | Residue (%) |
|---|---|---|---|---|
| TPC | 363.9 | N/A | 413.6 | 3.64140675 |
| AR-RH | 338.8 | 157.7 | 402.59 | 6.04912836 |
| H-RH | 351 | 157.8 | 403.72 | 6.69500997 |
| H-MSA-RH | 338 | 164.4 | 399.8 | 7.52431 |
| H-AL-RH | 337.7 | 171.8 | 403.75 | 5.80832 |

The derivative weight loss curve exhibits two distinct peaks, each corresponding to the decomposition of biomass and TPC, respectively (Figure 3 and Table 3). This

distinction is validated by the absence of peak 1 in the TPC sample. Peak 1 temperatures were comparable for AR-RH and H-RH composites, registering at 157.7 °C and 157.8 °C, respectively. Notably, the second-stage-treated fiber composites displayed higher peak one degradation temperatures, a favorable attribute for the FFF technique.

Peak 2, marked by substantial weight loss, represents the stage where the remaining lignin, hemicellulose, pectin, and cellulose undergo degradation alongside TPC depolymerization [19].

Analogous to the peak one temperature data, peak two temperatures demonstrated the highest value in the TPC sample at 413.6 °C, surpassing its composite counterparts.

These TGA outcomes offer comprehensive insights into the thermal behavior of pure TPC and rice husk composites. They highlight the impact of different composite compositions on the onset temperatures and degradation profiles, ultimately contributing to a better understanding of their thermal stability and suitability for additive manufacturing processes.

### 3.3. Rheology Study of Composites

The rheological characteristics of the composites were analyzed by studying their viscosities at different shear rates. As shown in Figure 4, the results revealed that all the composite variants exhibited shear-thinning behavior [20]. This means that the viscosities decreased as the shear rates increased. The viscosity measurement was performed at an optimized print temperature of 220 °C, which was discussed in Section 3.1.

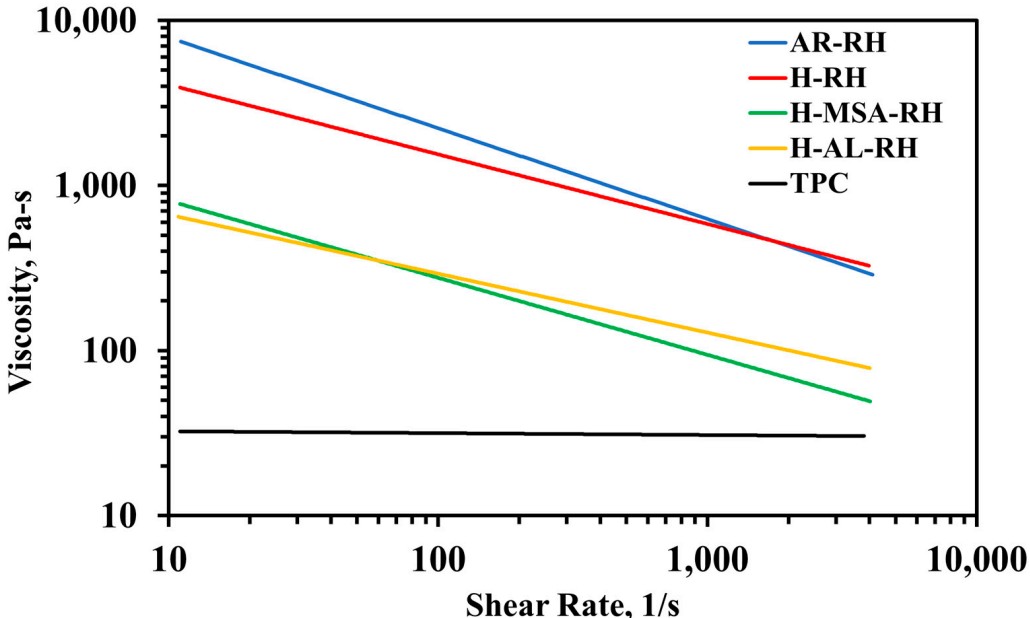

**Figure 4.** Viscosity at the print temperature of 220 °C with different shear rates.

The untreated and single-stage-treated rice husk composites showed higher viscosities overall, resulting in nozzle clogging and insufficient material flow. On the other hand, the two-stage-treated rice husk composites displayed lower viscosities at all shear rates. This improvement in flowability suggests a better potential for the distribution of fibers within the printed part. TPC had the lowest viscosity, which helped prevent nozzle clogging but could potentially compromise interlayer adhesion.

The steady-state torque and viscosity of the composites were evaluated at 160 °C to gain insights into their behavior during compounding and filament extrusion (Table 4). The results unveiled two distinct trends. One, the highest steady-state mixing torque was observed in pure TPC, recording at 45.29 N-m. Sequentially, the AR-RH composite exhibited a torque of 42.23 N-m, followed by H-AL-RH (39.07 N-m), H-MSA-RH (37.4 N-m), and H-RH (35.68 N-m). Second, similar to the mixing torque trend, the viscosity of the composites

at an extrusion temperature of 160 °C adhered to a similar trend. Pure TPC displayed the highest viscosity, succeeded by AR-RH, H-AL-RH, H-RH, and H-MSA-RH. The treated samples demonstrated lower viscosity and torque, indicating a more homogenous mixture during composite formulation and filament extrusion processes.

**Table 4.** Compounding steady state torque and viscosity at a filament extrusion temperature of 160 °C.

|  | Steady State Torque (N-m) | Viscosity at 160 °C |
|---|---|---|
| TPC | 45.29 | 5973.55 ± 292.58 |
| AR-RH | 42.23 | 4739.56 ± 178.87 |
| H-RH | 35.68 | 3403.44 ± 210.31 |
| H-MSA-RH | 37.40 | 3304.86 ± 256.29 |
| H-AL-RH | 39.07 | 4315.34 ± 283.88 |

These findings give us important information on how the composites behave rheologically and how this affects the printing process. The patterns observed emphasize the importance of the composition and treatment of the composite in influencing how the material behaves during additive manufacturing.

*3.4. Surface Morphology of Printed Composite Parts*

Optical microscopy imaging was performed on the top surfaces of the printed parts to assess surface roughness on a macro scale visually. Figure 5 presents the optical microscopy images, which provide valuable insights into the surface roughness characteristics of the printed parts.

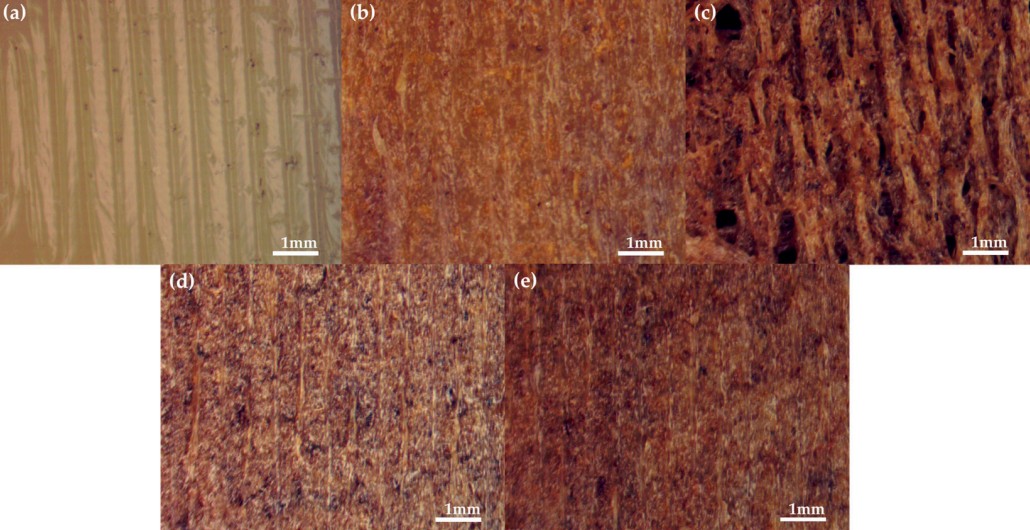

**Figure 5.** Optical microscopy images of top surface of 3D printed (**a**) TPC, (**b**) AR-RH, (**c**) H-RH, (**d**) H-MSA-RH, and (**e**) H-AL-RH.

Distinct surface features were observed for each composite, offering valuable information about their extrusion and printing behavior. The pure TPC displayed consistent material extrusion, but notable seams contributed to increased surface roughness. The AR-RH composite faced nozzle clogging challenges during 3D printing but exhibited comparatively subtle surface roughness. Both H-MSA-RH and H-AL-RH composites displayed even and smooth surfaces, characterized by minimal surface roughness, while the top surface of the H-RH composite lacked a smooth finish due to suboptimal material flow and nozzle clogging. These visual observations highlight the influence of composite composition and treatment on surface roughness and contribute to the comprehensive understanding of the printed parts' surface characteristics and quality.

SEM imaging shown in Figure 6 of the top surface, side profile, and cross-section of the printed parts was performed to observe the surface features, porosity, and overall homogeneity of the rice husk composites and the characteristics of rice husk composites. For pure TPC, the SEM images showed a smooth finish and clear layer-to-layer adhesion, indicating its desirable surface quality. However, fiber composite parts had a rougher surface finish yet displayed a more homogeneous appearance where distinctions between infill patterns were less discernible. The H-RH composite exhibited a significant amount of porosity and perimeter-infill gaps, making it stand out from other composites. The H-MSA-RH composite presented a relatively even top surface with tiny pores, which could be due to gas porosity during the printing process. Meanwhile, second-stage chemical pretreatment applied to H-MSA-RH and H-AL-RH composites appeared to have reduced porosity sizes and distribution. This outcome is potentially linked to removing hydroxyl groups, contributing to enhanced surface characteristics.

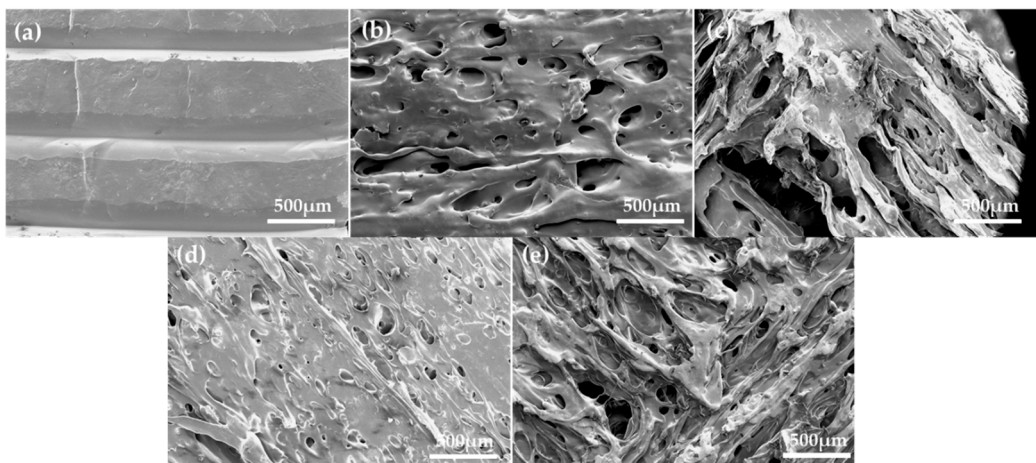

**Figure 6.** SEM of the top surface of printed parts using (**a**) TPC, (**b**) AR-RH, (**c**) H-RH, (**d**) H-MSA-RH, and (**e**) H-AL-RH.

Further SEM imaging was performed to understand the interlayer bonding quality and layer uniformity of the printed parts, revealing the influence of composite composition, treatment, and viscosity on additive manufacturing, as shown in Figure 7. A uniform layer thickness is evident for TPC, H-MSA-RH, and H-AL-RH composites, reflecting consistent and desirable interlayer bonding characteristics. AR-RH and H-RH composites exhibit layer delamination in the mid-section of the parts. This occurrence of uneven layer thickness may be attributed to the higher viscosities associated with AR-RH and H-RH, impacting material flow and interlayer adhesion. Porosity is noticeable in the side profile of the AR-RH composite; however, a significant reduction in porosity was observed after applying stage 1 and stage 2 pretreatments to the fiber composites. Moreover, additional SEM images were captured to understand the effectiveness of composite pretreatment strategies in optimizing the interfacial bonding characteristics in natural fiber-reinforced composites, as shown in Figure 8. Notably, a gap is apparent in the interfacial region of AR-RH and H-RH composites, suggesting suboptimal bonding. The average interface gap distance measures 6 μm for AR-RH and H-RH, contrasting with 1 μm for H-MSA-RH and H-AL-RH. The application of two stages of pretreatment leads to decreased hydrophilicity of rice husk fibers and tighter bonding with the TPC matrix. Enhanced bonding attributed to these pretreatments underscores the importance of effective fiber–polymer interfacial interaction for achieving high mechanical properties in natural fiber composites. Such bonding is crucial, due to the fibers' inherent hydrophilic nature and the polymer matrix's hydrophobic nature [21,22].

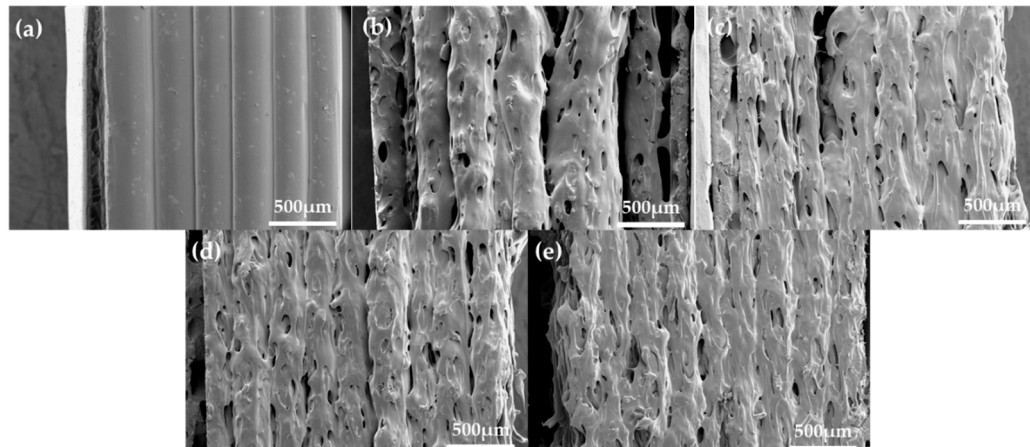

**Figure 7.** SEM of the side profile of printed parts using (**a**) TPC, (**b**) AR-RH, (**c**) H-RH, (**d**) H-MSA-RH, and (**e**) H-AL-RH.

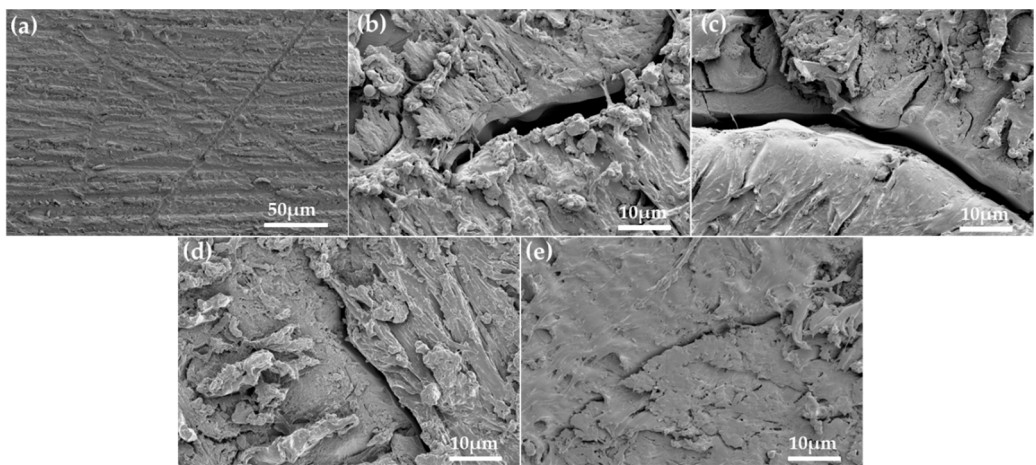

**Figure 8.** SEM of the cross-section of printed parts using (**a**) TPC, (**b**) AR-RH, (**c**) H-RH, (**d**) H-MSA-RH, and (**e**) H-AL-RH.

### 3.5. Mechanical Properties of Printed Composite Parts

Figure 9a displays the average filament diameters, offering insights into filament dimensional accuracy and consistency. The average filament diameters of all samples, except AR-RH, fall within the tolerance range suitable for a consumer 3D printer. The typical diameter commonly employed for 3D printing filaments is 1.75 ± 0.03 mm. The H-RH composite exhibits a higher standard deviation in filament diameter, potentially attributed to the presence of lignin and other components that may contribute to inconsistencies in filament dimension. The hydrophilic nature of untreated fibers could influence the larger average diameter of 1.85 mm observed in the H-RH composite. The increased moisture content in the fibers, particularly at the extrusion temperature of 160 °C, can convert moisture into water vapor. This process may contribute to porosity, filament swelling, and subsequent diameter enlargement. The presence of lignin and moisture-related effects on filament diameter highlights the need for comprehensive understanding and optimization in filament production processes [23]. Based on the data presented in Figure 9b, it appears that the H-MSA-RH composite has the highest hardness value among the different fiber composites. It is followed by AR-RH, TPC, H-RH, and H-AL-RH. However, it is worth noting that chemical pretreatment of the fibers tends to reduce the hardness values of treated fiber composites, with the exception of H-MSA-RH. Sulfuric acid and alkali treatments are known to remove hemicellulose and lignin, which makes the fibers softer and reduces their hardness. But for H-MSA-RH, the hardness actually increased by 10% compared to pure

TPC and 5% compared to untreated rice husk composites. It is possible that the unique structure of rice husks, which contains a high amount of silica and lignin, played a role in the structure of MSA-treated fiber composite [24].

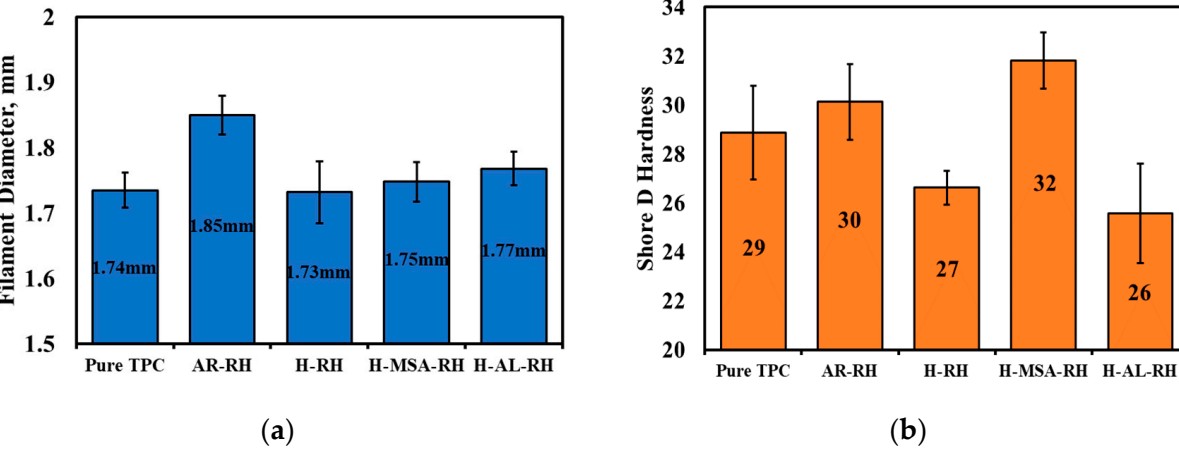

**Figure 9.** (**a**) Average filament diameter and (**b**) Shore D hardness.

Based on Table 5, it is evident that the stress at 5% and 50% strain and the elastic modulus for TPC and its rice husk composites are all important factors in determining the strength of the material. While some instances may make it impossible to generate yield stress due to TPC's elasticity preventing parts from yielding and fracturing, the stress at 5% and 50% strain can provide a good indication of the material's strength [16]. Notably, stress at 5% was similar for all the composites and TPC except for H-RH. Similarly, stress values were similar at 50% strain, with TPC exhibiting $6 \pm 1$ MPa while H-MSA-RH showed 5 MPa stress. Adding rice husk fibers decreased the elastic modulus from $26 \pm 4$ MPa to 21 MPa, a 19% decrease. However, the tensile test results showed that chemical pre-treatment improved the mechanical properties of rice husk-reinforced composites. This was evident in the 25% increase from 4 MPa to 5 MPa at 50% strain observed between AR-RH and H-MSA-RH. The elastic modulus also increased by 23.5% between AR-RH and H-MSA-RH composites from $17 \pm 5$ MPa to 21 MPa. These increases could be attributed to the material flow during printing, which led to decreased delamination and a better fiber–polymer interface. Although the mechanical property values were recorded to the nearest whole number and the error percentage was acceptable, it is worth noting that all the error was less than 20%, except for AR-RH and H-RH. This could be due to lower print quality and poor fiber–polymer bonding, which caused inconsistent fiber loading and higher standard deviation in mechanical property data.

**Table 5.** Stress at 5% and 50% strain and elastic modulus of TPC and rice husk composites.

|  | Stress at 5% Strain, MPa | Stress at 50% Strain, MPa | Elastic Modulus, MPa |
|---|---|---|---|
| TPC | 2 | $6 \pm 1$ | $26 \pm 4$ |
| AR-RH | 2 | $4 \pm 1$ | $17 \pm 5$ |
| H-RH | 1 | 3 | $13 \pm 2$ |
| H-MSA-RH | 2 | 5 | 21 |
| H-AL-RH | 2 | $5 \pm 1$ | $15 \pm 3$ |

## 4. Discussion

Our investigation provided valuable insights into the properties of chemically pre-treated rice husks when mixed with TPC polymer. Through the use of L9 Taguchi analysis, we determined the optimal print conditions for ensuring the accuracy and surface rough-ness of our printed parts. We measured various factors, including print temperature, print speed, and layer height, as these elements all play a crucial role in determining the con-

sistency and quality of the material being extruded by the nozzle. Our findings revealed that layer height significantly impacts the adhesion between layers, while print speed and temperature affect the consistency of the extruded material. By carefully selecting our print conditions, we ensured that our printed parts were of the highest quality possible. After conducting our analysis, we found that the best material extrusion occurred at a print speed range of 15–25 mm/s. Additionally, print temperatures ranging from 210–230 °C were closest to the operating temperature for TPC. Finally, layer height was selected based on preliminary data of TPC, and we found that a layer height of 0.25 mm, along with a print temperature of 220 °C and a print speed of 15 mm/s, provided the best print quality and reduced analysis errors due to poor printing [16].

Additionally, we found that the thermal stability of these materials is closely linked to their interfacial and mechanical properties. We conducted thermal stability analysis on the composites and pure TPC, ensuring the degradation temperature exceeded the printer operating temperature. While the addition of rice husks did cause a slight decrease in the degradation point of the composites, this change was not significant. We also investigated the shear-dependent viscosity of all samples at 220 °C to see how they behave while extruded by the 3D printer. Nozzle clogging was common, particularly in untreated and single-stage-treated rice husk composites [25].

The H-RH composite, in particular, showed poor top surface print quality. Interestingly, the shear-dependent viscosity of pure TPC almost had a slope of 0, which could be problematic because material dispersion on the print build plate may hinder layer-to-layer adhesion. However, the second-stage pretreated rice husk composites had lower viscosity and mixing torque. This indicates the material is less energy-intensive and achieves a better fiber–polymer interface during filament extrusion at 160 °C. This was reflected in surface morphology studies, where the second-stage pretreated fibers had a better interface with the polymer matrix. Our analysis was highly repeatable, with an average error percentage of about 5%, falling within an acceptable range.

## 5. Conclusions

Rheological analysis revealed that the two-stage-treated rice husks demonstrated improved viscosity profiles, resulting in consistent material extrusion and enhanced surface quality in printed parts. The mechanical properties indicated that the addition of chemically pretreated rice husk fibers improved the tensile properties of the composites, with the H-MSA-RH composite showcasing notable enhancements in tensile stress at 50% strain and Young's Modulus. These improvements were attributed to the defibrillation of fibers, leading to enhanced interfacial bonding and overall material strength. Interestingly, the hardness values of treated fiber composites, except H-MSA-RH, were reduced, due to the removal of hemicellulose and lignin during chemical pretreatment. This suggests that the material's mechanical properties can be tuned based on specific requirements by adjusting the pretreatment methods.

Furthermore, the optical microscopy and SEM imaging revealed distinct surface characteristics for each composite, shedding light on the printed part's extrusion behavior and surface roughness and applying two-stage pretreatments, such as the MSA treatment, improved surface quality, reduced surface roughness, and enhanced overall part aesthetics. The SEM images further elucidated the printed parts' interlayer bonding quality and porosity distribution. This comprehensive examination highlighted the influence of composite composition, treatment, and viscosity on additive manufacturing outcomes.

The successful integration of two-stage chemically pretreated rice husk fibers, particularly the H-MSA-RH composite, into the TPC matrix demonstrates the potential to achieve desirable material properties while reducing environmental impact. This aligns with the growing emphasis on sustainable materials in additive manufacturing and contributes to developing eco-friendly composite solutions. We have demonstrated how advanced pretreatment techniques can enhance material properties by integrating chemically pretreated rice husk fibers into NFC filaments. These insights pave the way for developing

resilient, cost-effective, and eco-friendly composite materials that can be used across various industries, from medical to aerospace and automotive.

**Author Contributions:** Conceptualization, A.N.S. and K.K.; methodology, A.N.S. and S.M.; investigation, A.N.S. and S.M.; writing—original draft preparation, A.N.S.; writing—review and editing, A.N.S. and K.K.; supervision, K.K. and J.S.; project administration, K.K. and J.S.; funding acquisition, K.K. and J.S. All authors have read and agreed to the published version of the manuscript.

**Funding:** This research was funded by United States Department of Agriculture, grant number 2021-67021-34768.

**Data Availability Statement:** The authors are not able to provide datasets publicly for privacy reasons. Interested researchers can contact corresponding author if datasets are needed.

**Acknowledgments:** The authors would like to acknowledge Micro/Nano Technology Center, University of Louisville (MNTC) for their help in operating the SEM. The authors would also like to thank Conn Center for Renewable Energy for characterization facilities.

**Conflicts of Interest:** The authors declare no conflict of interest. The funders had no role in the design of the study; in the collection, analyses, or interpretation of data; in the writing of the manuscript; or in the decision to publish the results.

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
