# Peer review of "Influence of Chemical Pretreatment on the Mechanical, Chemical, and Interfacial Properties of 3D-Printed, Rice-Husk-Fiber-Reinforced Composites"

_jcs, doi:10.3390/jcs7090357_

Round 1

Reviewer 1 Report

The submitted manuscript cannot be accepted for publication in this form, but it has a chance of acceptance after a major revision. My comments and suggestions are as follows:

 1- Abstract gives information on the main feature of the performed study, but a couple of sentences about the details of conducted tests must be added.

2- Authors must clarify necessity of the performed research. Research questions, aims and objectives of the study must be clearly mentioned in introduction.

3- The literature study must be enriched. In this respect, authors must read and refer to the following relevant papers: (a) https://doi.org/10.3390/polym13101559 (b) https://doi.org/10.1016/j.apsusc.2021.149602 and other research works.

4- It is necessary to add limitations and advantages of the study.

5- Why this particular materials are considered for the asphalt mixture in this study.

6- Introduction is too short in its current version. The relevant studies must be critically reviewed and compared.

7- Why these particular parameters (Print Speed, Layer Height, Print Temperature) and these values (Table 1) are considered in this study.

8- For "Surface Morphology of Printed Parts", authors must add figures from microscope to show surface roughness and results of chemical treatment.

8- Novelty needs to be explicit in highlights and abstract. Also, images must be more informative.

9- Repeatability in the obtained results must be explained and discussed.

10- Standard deviation in the presented curves must be discussed. In addition, error in calculation must be discussed.

11- The conclusion must be more than just a summary of the manuscript. List of references must be updated based on the proposed papers. Please provide all changes by red color in the revised version

Minor editing of English language required.

Author Response

The authors would like to thank the reviewer for their comments and suggestions. The objective of this study is to investigate the effects of different chemical pretreatment methods to improve the interface between the fibers and polymer matrix which in return improves other properties such as mechanical, thermal and rheological. Pretreatment of rice husks prepared the fibers for 3D printing applications with natural fiber composite filament fabrication.

  • Abstract gives information on the main feature of the performed study, but a couple of sentences about the details of conducted tests must be added.

The authors added relevant information in lines 16-20.

  • Authors must clarify necessity of the performed research. Research questions, aims and objectives of the study must be clearly mentioned in introduction.

Novelty and aims for this study are mentioned in lines 88-92. The objective of the study was added in lines 102-104.

  • The literature study must be enriched. In this respect, authors must read and refer to the following relevant papers: (a) https://doi.org/10.3390/polym13101559 (b) https://doi.org/10.1016/j.apsusc.2021.149602 and other research works.

The authors would like to thank the reviewer for these two literature sources. The authors have included this information in the introduction.

  • It is necessary to add limitations and advantages of the study.

Limitations and advantages of this study are mentioned in the introduction (lines 93-96) and discussion (lines 386-389).

  • Why these particular materials are considered for the asphalt mixture in this study.

The authors are not investigating asphalt mixture. We are using thermoplastic co-polyester, which is an elastomer typically used in automotive interiors. Rice husks are chosen for this study because of their composition (high lignin and silica content), large volume availability, and to provide a value-added application for rice husks which is currently burnt for energy.

  • Introduction is too short in its current version. The relevant studies must be critically reviewed and compared.

Thanks for the suggestion. The authors have added more studies to the introduction per your suggestion.

  • Why these particular parameters (Print Speed, Layer Height, Print Temperature) and these values (Table 1) are considered in this study.

Typically, these parameters contribute the most to the print quality of any parts, whether it is a composite or pure material. Layer height affects layer to layer adhesion, while print speed and temperature affects the consistency and of the material being extruded by the nozzle. This is added in the manuscript (lines 351-358).

  • For "Surface Morphology of Printed Parts", authors must add figures from microscope to show surface roughness and results of chemical treatment.

The authors added this in the results section under surface morphology (Figure 5).

  • Novelty needs to be explicit in highlights and abstract. Also, images must be more informative.

The last sentence of the abstract shows the novelty of this study. The authors have showed that a 2-stage acid treatment of rice hulls leads to improved polymer-fiber adhesion in the composites and improved elongational properties. Previous studies reported in the literature were done using alkaline treatment for pretreatment of rice husk but not the 2-stage acid treatment.

  • Repeatability in the obtained results must be explained and discussed.

Repeatability is discussed in the discussion section (lines 341-345, 376-377).

  • Standard deviation in the presented curves must be discussed. In addition, error in calculation must be discussed.

Standard deviation and errors are discussed in the results and also the discussion sections. (line 311-313, 343-345).

  • The conclusion must be more than just a summary of the manuscript. List of references must be updated based on the proposed papers. Please provide all changes by red color in the revised version

The authors have included discussions and comparison to literature, as well as how the results bring the analysis together and relate to each other.

Reviewer 2 Report

At the faculty of the university where I work, there are research teams involved in the study of polymer composites with organic additives. Recently, they have been conducting extensive research on biodegradable polymers with wood, corn, coffee, etc. Therefore, I think that this is current and important research, and it would be interesting for me to determine the production technology of such a composite and the impact on the quality of filament for 3D printing. The authors focused on the quality of the composite, presented the results of their research, which show an interesting scientific achievement.

Author Response

The authors would like to thank the reviewer for their comments

Reviewer 3 Report

The manuscript entitled ‘Influence of Chemical Pretreatment on the Mechanical, Chemical, and Interfacial Properties of 3D Printed Rice Husks Fiber reinforced Compositesis” in line with the Journal of Composites Science. This article is based on original research. The topic of the article is connected with of application the composites in a new technology, additive manufacturing. It is up-to-date and important. The manuscript is well composed; nevertheless, it requires a careful edition and some other changes before publication, including:

·       Abstract: check lines 13 and 14 there is two time “second-stage”.

·       Abstract: add research methods.

·       Introduction: the number of cited references should be directly after the author’s name.

·       Introduction: consider also this article as a source of information about natural fiber modification methods: DOI: 10.1051/matecconf/202032201012

·       Chapter 2: please correct the sub-chapters numbers it should be “2.1.” etc instead of “3.1” etc.

·       Chapter 3.1: rice husk required more detailed characteristic physical and chemical properties, including size…

·       Chapter 3.2: line 99 and many other places in this manuscript – error of source. It requires modification.

·       Chapter 2: add the table with the designation of samples.

·       Figures: requires to be order, including reference in the text. There are two time figures 1 and 2 in the text. Not all references in the text to the figures are correctly pointed out.

·       Discussion: the chapter named “Discussion” has a character of conclusion part. There is a lack of proper discussion and comparison with the literature in the article.

Minor editing of English language required

Author Response

The authors would like to thank the reviewer for their comments and suggestions. The objective of this study is to investigate the effects of different chemical pretreatment methods to improve the interface between the fibers and polymer matrix which in return improves other properties such as mechanical, thermal and rheological. Pretreatment of rice husks prepared the fibers for 3D printing applications with natural fiber composite filament fabrication.

  1. Abstract: check lines 13 and 14 there is two time “second-stage”.

The reason for having “second-stage” twice is to make it clear to the audience that both MSA and alkali are two stage processes, where first stage was using sulfuric acid and second stage was either MSA or sodium hydroxide. The authors reworded this for better understanding.

  1. Abstract: add research methods.

This has been added to the abstract lines 16-20.

  1. Introduction: the number of cited references should be directly after the author’s name.

The authors updated the citations to immediately after the author’s name.

  1. Introduction: consider also this article as a source of information about natural fiber modification methods: DOI: 10.1051/matecconf/202032201012

We thank the reviewer for this article. We have added in our introduction.

  1. Chapter 2: please correct the sub-chapters numbers it should be “2.1.” etc instead of “3.1” etc.

The authors have corrected this, thank you

  1. Chapter 3.1: rice husk required more detailed characteristic physical and chemical properties, including size…

The current study is an extension of another study which focuses more on the chemical pretreatments of rice husks. The authors added some physical characteristics of the rice husks used in the methodology section 2.1.

  1. Chapter 3.2: line 99 and many other places in this manuscript – error of source. It requires modification.

The authors have corrected this. It seems like there was an error while converting files. Thank you.

  1. Chapter 2: add the table with the designation of samples.

The authors have added that in section 2.1.

  1. Figures: requires to be order, including reference in the text. There are two time figures 1 and 2 in the text. Not all references in the text to the figures are correctly pointed out.

The authors have corrected this.

  1. Discussion: the chapter named “Discussion” has a character of conclusion part. There is a lack of proper discussion and comparison with the literature in the article.

The authors have included some discussions and comparison to literature in this section.

Round 2

Reviewer 1 Report

The paper has been improved and corresponding modifications have been conducted. In my opinion, the current version can be considered for publication.

Reviewer 3 Report

The manuscript entitled ‘Influence of Chemical Pretreatment on the Mechanical, Chemical, and Interfacial Properties of 3D Printed Rice Husks Fiber reinforced Compositesis” has been corrected according given suggestions. It is ready to final editing by journal.

 Minor editing of English language required.